# The Influence of Graphene in Improvement of Physico-Mechanical Properties in PMMA Denture Base Resins

**DOI:** 10.3390/ma12142335

**Published:** 2019-07-23

**Authors:** Cecilia Bacali, Mindra Badea, Marioara Moldovan, Codruta Sarosi, Vivi Nastase, Ioana Baldea, Radu Stefan Chiorean, Mariana Constantiniuc

**Affiliations:** 1Department of Prosthodontics and Dental Materials, Iuliu Hatieganu University of Medicine and Pharmacy, 32 Clinicilor, 400006 Cluj-Napoca, Romania; 2Department of Preventive Dental Medicine, Iuliu Hatieganu University of Medicine and Pharmacy, 31 Avram Iancu, 400083 Cluj-Napoca, Romania; 3Department of Polymer Composites, Institute of Chemistry Raluca Ripan, Babes-Bolyai University, 30 Fantanele Str., 400294 Cluj-Napoca, Romania; 4Institute for Computational Linguistics, University of Heidelberg, 325 Im Neuenheimerfeld, 69120 Heidelberg, Germany; 5Department of Physiology, Iuliu Hatieganu University of Medicine and Pharmacy, 1 Clinicilor, 400006 Cluj-Napoca, Romania; 6Department of Mechanical Engineering, Faculty of Mechanics, Technical University of Cluj-Napoca, 103-105 Muncii Bld., 400641 Cluj-Napoca, Romania

**Keywords:** denture, graphene-Ag nanoparticles, polymethyl methacrylate, mechanical characterization, water absorption

## Abstract

The clinical performances of dental materials depend on their mechanical profiles, determining their long-term deformation and wear resistance. This paper describes a study on the mechanical properties, water absorption and morphological properties of a polymethyl methacrylate (PMMA) resin enriched with graphene-silver nanoparticles (Gr-Ag). Two different concentrations—1 and 2 wt.%—of Gr-Ag were loaded into the PMMA material. For the mechanical characterization, the compression behavior, flexural strength and tensile strength were evaluated. Optical microscopy in polarized light and scanning electron microscopy were used for filler analysis. The filler addition led to an improvement in all mechanical properties, with slight changes being derived from the filler content variation. Gr-Ag use led to an increase in the applied maximum loads. Moreover, 1 wt.% Gr-Ag determined an increase of 174% in the modulus of rupture, which indicates high flexibility.

## 1. Introduction

Biomaterial science and engineering research have made important progress over the last few years, due to their contributions in health-related domains [1]. Several domains, such as restorative medicine and prosthetic dentistry, benefit from the expanding research due to the constant need for biocompatible non-toxic materials with good mechanical properties and high wear resistance [2].

Restoration of oral and masticatory function using a removable prosthesis is essential to improving the quality of life of a continuously aging population [3]. However, a frequent clinical issue reported worldwide is high denture fracturability. In the year 1997, more than one million denture repairs were necessary in the UK to overcome the low fracture and impact strength of denture resins. Apparently, close to 70% of dentures break in the first three years of wearing due to repeated masticatory forces causing material fatigue, or as a result of prosthesis dropping. Moreover, the stresses developed in the oral cavity during biting and mastication have a deforming effect, which can initiate fractures in time [4].

For more than 80 years, dentures have been made from polymeric materials such as polymethyl methacrylate (PMMA), and recently, for non-metal clasp dentures (NMCD), polyamide, polyester or polycarbonate injection molded thermoplastic have been used. PMMA is commonly used due to its high availability, low cost, acceptable aesthetics, and versatility with respect to shaping and biocompatibility [5]. Some of its biological properties can even be improved by chemical tuning. Reducing biofilm formation on dental materials is essential to oral health, and so is developing alternative therapies to the current use of antibiotics for drug resistant bacteria. Tooth is end, various additives such as zirconium oxide nanoparticles, silver nanoparticles and platinum nanoparticles were incorporated into acrylic resins to reduce bacterial and fungal colonization, with good results on *Candida albicans* adhesion and antibacterial activity [6]. Baker et al. demonstrated the antibacterial properties of silver nanoparticles of different sizes at low concentrations both in solution and on Petri dishes; he found that smaller particles could be more efficient against bacteria due to their large surface to volume ratio [7]. Loading silver nanoparticles into lactose-modified chitosan for heat-polymerized PMMA showed important antibacterial results on both gram + and gram − strains [6]. However, some of the materials showed poor mechanical strength, low surface hardness and brittleness. Moreover, loading silver nanoparticles produced different effects on the PMMA matrices, depending on the curing method: microwave polymerized resins showed high flexural strength and elastic modulus, while heat-polymerized resins had poor flexural profiles [8].

The mechanical properties of dental materials play a crucial role in their clinical performance and are strongly related to both processing and composition. For example, complete dentures, used for esthetic and functional restoration in edentulous patients, are subjected to high and complex mastication forces, that can lead to fractures. Improving resistance by increased denture thickness can be troublesome for patients, hence the importance of exploring other possibilities to enhance the mechanical properties of the acrylic resin.

Additionally, dentures are used inside the oral cavity, in a wet environment that can lead to dimensional changes and affect the resin’s characteristics as water molecules enter between the polymeric chains and act as plasticizers, altering the physical and mechanical properties of the material; therefore water absorption of new denture base resins is important to be studied [9]. 

PMMA is known for easy preparation, but polymerization shrinkage and distortion of the denture base were reported [10]. At the same time, wet storage determines a slight dimensional increase that could compensate for the polymerization shrinkage. The increase in curing temperature was reported to yield pore-free structures and low levels of residual monomer associated with low water absorption [11]. NMCD materials such as polyamide, polyester or polycarbonate showed better fracture- and impact-resistance than acrylic materials, but they were difficult to grind, had low color stability, and needed short span maintenance [12]. Therefore, extensive research was carried out to improve the properties of acrylic denture base materials by mixing PMMA with suitable fillers. The incorporation of 2% zirconium oxide, as well as rubber (poly n-butyl acrylate—PBA and nitrile butadiene rubber) addition associated with ceramic fillers [4], showed an improvement in impact strength. Acrylic resins enriched with 0.25% or 0.5% pristine nanoclay improved flexural strength, flexural modulus and fracture toughness [13]. Polyvinylalcohol electro spun nanofibers had a promising effect on PMMA flexibility and overall mechanical performance [14]. 

Graphene, known since 2004 as “the thinnest material in the universe”, has attracted great interest in diverse research fields including dentistry for its unique mechanical properties [15]. As a carbon allotrope, it consists of a one-atom-thick sheet of sp^2^ bonded carbon atoms densely packed in a honeycomb crystal structure [6,15]. Graphene oxide (GO), as well as reduced graphene oxide (rGO), can be easily functionalized due to their abundance in oxygen containing groups (epoxy, hydroxyl and carboxyl),enabling their use in a series of nanocomposites of different polymers with a wide range of applications [16,17,18,19].

In an attempt to improve both mechanical denture composite properties, PMMA resin was loaded with graphene-Ag nanoparticles (Gr-Ag). The present study aimed at the characterization of the new composites obtained by loading different ratios of Gr-Ag into a commercial auto-polymerized PMMA resin. The samples were evaluated for the compression behavior, flexural properties and tensile strength, as well as by their water absorption profiles and morphological aspects.

## 2. Materials and Methods

### 2.1. Graphene-Based Nanomaterial Preparation

Graphene-silver nanoparticles (Gr-Ag) used in this study were synthesized (in the Synthesis and Applications of Carbon Nanostructures Department of Institute for Research and Development of Isotopic and Molecular Technologies, Cluj-Napoca, Romania) by radio-frequency chemical vapor deposition (RF-CCVD) using catalytic silver nanoparticles distributed over magnesium oxide (Agx/MgO, where x = 3 wt.%). The synthesis was carried out using a methane flow of 80 mL/min and a reaction time of 60 min [20].The obtained graphene had6 layers decorated with Ag nanoclusters.

The experimental materials of a liquid self-polymerizing denture acrylic were obtained from Castavaria (Vertex Dental B.V., Centurion Baan 190, 3769 AV, Soesterberg, The Netherlands). The control sample contained only the acrylic denture material (M), while the samples P1 and P2 contained 1 (P1) and 2 (P2) wt.% Gr-Ag. The Gr-Ag obtained using a method described by Sava et al. [21], were mixed in the PMMA powder, using a vibrating lab device, then the powder and liquid mixture was cured for 30 min, at 55°C and 2.5 bar in a polymerizing device, as described in the product specifications. Sticks of 150mm/5mm were obtained, which were then sectioned in 1.5 mm disks, 10 for each sample. The sticks obtained are presented in Figure 1.

### 2.2. Mineral Glass Optic Microscopy in Polarized Light with Crossed Nicols

The polymer powder was stretched out on the glass blade and investigated using Optical mineralogic microscope with cross polarized light, Laboval 2 (Carl Zeiss Jena, Oberkochen, Germany), with an 8-megapixel Samsung digital camera (Samsung, Seoul, Korea).

### 2.3. Mechanical Property Evaluation

The samples for the mechanical evaluation of compressive strength (CS), diametral tensile strength (DTS) and flexural strength (FS) were prepared according to ISO 4049/2000 and international norms “American Dental Association’s Specification” No. 27 [21]. Silicone molds were used for preparation, to minimize the formation of cracks and flaws within the material. Ten samples of predetermined shapes and sizes were prepared for each test group, in accordance withthe designated testing method: 6 mm × 3 mm for DTS, 4 mm × 8 mm for CS and 2 mm × 2 mm × 25 mm for FS. After 24 h, samples were evaluated using a Lloyd Instruments-LR5k (LF Plus, LLOYD, Instrument, Ametek Inc., West Sussex, UK) mechanical testing machine equipped with Nexygen Software (version 3.0, Lloyd Instruments Ltd. Steyning Way, Bognor Regis, West Sussex, UK). The statistical significance of the values of the 10 samples for each of the three materials was tested using a one-way ANOVA (Microsoft Office, Excel, Python 3.5 and Matplotlib, San Diego, CA, USA).

### 2.4. Water Absorption

For the determination of water absorption and the absorption of artificial saliva, a silicone mold was used to obtain disc shaped specimens of 15 ± 1 mm diameter and a thickness of 1 mm. The discs were removed from the mold and dried in the desiccator in the presence of calcium chloride at 37°C for 24 h to obtain a constant weight (m_0_). They were divided into two groups: the samples from the first group were immersed in distilled water, while the samples from the second group were maintained inartificial saliva and placed in an incubator at a constant temperature of 37°C for 28 days. Every day, the samples were removed from the water and weighed after 10 min (m_1_) using an electronic analytical balance. The recorded weight changes were calculated using Equation (1) [22].
Wa = (m_1_ − m_0_) × 100/m_0_(1)

The statistical significance of the differences in weight for the 10 samples for each of the three materials was tested using a one-way ANOVA test (Microsoft Office, Excel).

### 2.5. Scanning Electron Microscopy

The scanning electron microscopy of the fillers and the samples (discs) was carried out using an FEI Inspect microscope (SEM-Inspect S, FEI Company, Hillsboro, OR, USA), S model, functional in high-vacuum and low-vacuum, tan accelerating voltage between 200V and 30kV. The microscope is equipped with CCD-IR infrared inspection camera and backscatter electron detector, with image processing up to 4096×3536 pixels. Typically, the images were collected with a magnification of 500 and 1000 times.

## 3. Results

### 3.1. Optical Microscopy in Polarized Light

The optical images of microscopy in polarized light with crossed nicols on the powders from the composition of the investigated materials are presented in Figure 2.

The polymer particles of PMMA (Figure 2a) have a spherical shape with diameters 25–50 μm. Under microscopic observation in polarized light with crossed Nicols, they have a “ghostly look”, with white edges and a dark interior with a white halo. The Gr-Ag powder (Figure 2b) is very fine, with a high tendency for coalescence between the particles. The predominant aspect of the powder is dark under microscopic observation in polarized light with crossed Nicols, which can be correlated with a structure with low crystallinity of the powder. In addition to the dark areas, there are very fine yellow-brown dots that correspond to silver. It shows crystalline features, enabling the particles to be pulverized in the pulverulent mass. The diameter of these particles is very small, the most representative of them having only a diameter of 2.5 μm. The incorporation of the Gr-Ag powder into the mass of the powdered polymer material (Figure 2c) takes place by fixing them to some of the polymer spheres. At high magnification, in polarized light with crossed Nicols, we observe spheres of compact black shade with shining spots in which we find yellow shades. These are the functional graphene spheres. The points of silver predominantly have diameters less than 3 μm. The spheres which did not get the functional material retain their whitish-ghostly appearance with the more obvious white edges.

### 3.2. SEM Analyses of Fillers

The PMMA, Gr-Ag and PMMA+1%Gr-Ag fillers were analyzed by scanning electron microscopy, as presented in Figure 3a–c.

The images of PMMA particles (Figure 3a) were spherical and the average particle diameter measured was about 23–91 μm. By analyzing the SEM images of the Gr-Ag filler (Figure 3b), an agglomeration of the particles is presented, but a specific morphology of the wrinkled graphene with few layers and irregular shape can be observed. According to [20], the white spots visible on the surface of graphene correspond to the Ag nanoparticles.

It can be observed in Figure 3c that Gr-Ag are dispersed on the surface of the PMMA particles, evidenced by the arrows.

### 3.3. Compression

The compression test results are presented in Figure 4. They revealed that the compression parameters were significantly influenced by the loading of Gr-Ag. The maximum load applied on the M sample was of 1614.3 ± 84.5 N, with a stress of 66.6 ± 4.4 MPa. Samples P1 and P2 displayed increasing values of stress at maximum load, which were correlated with the Gr-Ag ratio, up to 80.4 ± 5.4 MPa at a content of 2% Gr-Ag. The stress at rupture for sample M was of 59.2 ± 4.4 MPa at a strain of 36.4%. Data show that the addition of 1% Gr-Ag yields a similar stress at rupture at higher strain than the M sample, while 2% Gr-Ag leads to a stress increase of up to 64.3 ± 4.3 MPa at the same strain as sample M.

### 3.4. Flexural Strength

As shown in Table 1, the maximum load attained during the bending test varied significantly between the tested samples. The M sample containing only PMMA withstood 18.1 ± 4.2N, while P1 and P2 samples were exposed to 53.8 ± 19.2 N and respectively 81.3 ± 18.3 N. The modulus of rupture increased significantly from 13.4 ± 3.1 MPa for sample M to 35.0 ± 8.3 MPa for sample P1, while between the samples P1 and P2 a slight increase was obtained; however, this had no statistical significance.

### 3.5. Tensile Strength

Figure 5 displays the tensile strength evaluation results obtained for the three tested samples. The maximum load attained during the tensile strength test increased from 371.5 ± 81.2 for sample M to 535.3 ± 46.1 for sample P2. A significant difference (*p* < 0.05) was obtained between the samples P2 and M, and therefore for 2%Gr-Ag loading. The same variation was observed for the tensile stress at maximum load, with the overall increase of values, but with significant changes only for P2 samples, compared to the M sample. An increase was also shown for the Modulus, from 17.7 ± 4.1 MPa for sample M to 28.7 ± 7.4 MPa for sample P2; however, without statistical significance (*p* > 0.05) between P1 and P2.

Different superscript letters indicate statistically significant variation (*p* < 0.05) between the samples. For a better interpretation of the results, the experimental stress versus strain curves of samples M, P1 and P2 were plotted. As shown in Figure 6, the tensile properties of sample P1 is higher than for sample M, with the highest stress recorded for sample P2. The M samples fail at a strain of 2.3%, while the P2 sample fails at 3.2%, under a maximum stress of 44.6 ± 3.8 MPa.

### 3.6. Water Absorption

A series of samples was kept for 28 days in distilled water; another series was exposed to simulated saliva fluid, to mimic clinical conditions. The samples were monitored for 28 days and their weight increased gradually up to 20–25 days, when they reached a constant value. The absorbed water values exhibited an increased percentage after 7 and 28 days of water/saliva exposure, as presented in Table 2.

A significant increase in absorption was obtained for all samples from the 7th day of exposure until the 28^th^ day in both mediums. In the group exposed to simulated saliva fluid, the highest water absorption values were obtained for the control sample M and a decrease was recorded for the samples with Gr-Ag loading. For the samples evaluated after 7 days of exposure, the values decreased significantly at the loading of both 1% and 2%, with important differences at higher Gr-Ag content. However, the constant absorption values attained after a longer exposure time revealed no significant variation between the samples P1 and P2. The same behavior was recorded also for the samples exposed in distilled water, but the values were slightly higher.

### 3.7. SEM Analyses

The morphology of the samples’ surfaces was examined by electron microscopy and representative images captured at a magnification degree of 500× and 5000× are shown in Figure 7. For all the samples, the areas (pointed to by the arrows) corresponding to the pre-polymerized PMMA beads are visible. The image depicting sample P1 (Figure 7b) shows white areas at the perimeter of the PMMA spheres, with the largest surfaces for sample P2 (Figure 7c), indicating the deposition of the filler in the PMMA matrix between the spheres. Figure 7d shows a detail of a plain, uncovered PMMA sphere in sample M, while in Figure 7e, filler particles are distributed around the PMMA beads and in Figure 7f the spheres seam completely covered by the particles.

The fracture surface morphologies of samples M, P1 and P2 are depicted in Figure 8. The fracture surface of sample M displayed a squamous-like morphology with characteristic brittle structure with visible tracks of cracking in sharp edges. The samples reinforced with Gr-Ag showed smoother surfaces, with more ductile cracks and homogenously distributed particles in the PMMA matrix. 

## 4. Discussion

Acrylic resins are polymers of methacrylic acid esters with an alkyl group. PMMA is the most representative, also known as “organic glass” for its high levels of transparency. It exhibits excellent heat resistance, high strength and rigidity at low costs, but also important brittleness and poor impact resistance [12]. A plethora of studies focused on overcoming the disadvantages of PMMA denture resins. Previous research has reported the use of several fillers to reinforce the PMMA structures, such as: zirconium oxide, rubber derivatives, nanoclay, glass fiber and graphene [22,23,24,25]. Data showed that the presence of the high-elastic modulus graphene in a low modulus polymeric matrix can lead to important reinforcement; therefore, many studies have approached the mechanical properties of graphene nanocomposites [26]. Graphene loading varies in the literature from 0.5 to 20% [21,27,28]; however, the best results were reported for low filler contents [28], which determined the preparation of the two PMMA composites in the present study, containing 1% and 2% Gr-Ag.

The resistance of the material to the stresses caused by mastication can be predicted by evaluating its mechanical properties. To investigate the mechanical performance of graphene-based PMMA composites, compression measurements were performed and the stress (MPa) vs. strain (%) profiles were plotted. To get reliable data, ten samples were tested for each of the products and the mean stress at rupture (MPa), strain % and Young’s modulus were calculated together with the corresponding standard deviations.

Generally, the addition of nanofillers to polymer matrices stiffens the product, with a strain at rupture decrease and an enhancement of tensile strength [29]. The work of rupture obtained in this study varied in the same manner as the strain at rupture, meaning that P1 sample recorded an increase of work at rupture, while the value for sample P2 slightly decreased. Literature reports mentioned the association between work of fracture decrease and the presence of aggregates that function as stress raisers and induce early fractures [30]. Tripathi et al. showed that the addition of GO beyond 1% leads to the deterioration of mechanical strength attributed to the filler agglomeration effect and poor stress transfer characteristics [31]. In the present study, PMMA exhibits a stress at maximum load of66.66 MPa and in the graphene composites it gradually increases with the increase of Gr-Ag ratio. P1 sample withstands 13% higher stress, a significant increase compared to sample M containing PMMA. The addition of graphene structures determines strong chemical interactions in the matrices and their homogenous dispersion generates uniform stress distribution, leading to an important increase in mechanical strength of the composites [21,22]. Previous reports state that the improvement in the mechanical properties of polymers combined with graphene is determined by the interfacial interaction of the graphene with the polymer matrix and by the efficient transfer of the exterior load from the polymer to the filler [4,32]. Although the graphene loading was rather low, at 1 and 2%, it improved the compression parameters especially at 1% loading.

One of the major concerns regarding PMMA dentures relates to their flexural properties that are associated with the wear resistance and robustness. Materials with low flexural strength are prone to fractures and cracks during denture use and do not withstand continuous mastication-induced deformation [33]. Flexural properties were assessed by the triple point bending test in which a load is applied on a rectangular sample and as a result it develops a wide range of stresses across its depth. On the concave face, the stress will be at its maximum compressive value, on the convex face the stress will be at its maximum tensile value, while the extreme edges of the sample will be exposed to shear stresses. The denture composites are exposed to clinical conditions that combine all those complex stresses; therefore, the flexural strength analysis is an important evaluation criterion.

The modulus of rupture also known as flexural strength represents the highest stress experienced within the material at the moment of its rupture and describes the material’s ability to resist deformation under a certain load [34]. Shakeri et al. reported a 32% improvement in flexural strength when loading 0.05% Ag nanoparticles to PMMA denture bases [13], but without the graphene addition. In our case, adding 2% Gr-Ag to the PMMA resin led to an increase of 174% in the modulus of rupture, which could be due to the higher filler percentages and to the reinforcement properties of graphene.

Concerning the tensile strength testing, it can be clearly deduced that the loading of 2% Gr-Ag has completely changed the behavior and the mechanical properties of PMMA resin. Most of the evaluated mechanical parameters were positively influenced by the addition of 1% Gr-Ag; however, the increase in Gr-Ag loading induced significant improvement only to a few of the tensile strength indicators. Although Ag nanoparticles proved to exhibit antibacterial effects, a decrease in the mechanical properties was reported by different authors [35,36], so we decided to add graphene in the attempt to improve the mechanical properties as well.

The soluble component of the acrylic polymerized resin is the residual monomer, the uncovered methyl methacrylate (MMA) in the polymerization reaction, which is incriminated, together with its hydrolysis product, the methacrylic acid, in PMMA cytotoxic effects and adverse reactions [37]. Therefore, for chemical polymerizing resins, the water absorption seems to be related to the curing method due to the different amounts of MMA yielded. High curing temperatures often lead to high density structures, with low residual monomer content and low water absorption [11]. Previous reports on the reinforcement filler’s influence on residual monomer content showed that graphene oxide can act as a polymerization inhibitor by premature chain termination, thus lead to important ratios of residual monomer [38] and further to high water absorption. However, the association of PMMA with carbon-graphite fiber decreased water absorption at high fiber loading, due to the increase in the ratio of insoluble composites [39]. Our results were in good agreement with the previously mentioned reports, showing a decrease in water absorption when Gr-Ag loading increased, probably due to a better filler distribution and a lower porosity. When the final constant absorption capacity was reached, our results showed a significant decrease of absorption for sample P1 with 1% Gr-Ag loading compared to sample M and a slight decrease although with no statistical significance for sample P2. These results could be correlated with the mechanical characterization that showed the same behavior with significant improvement for 1% Gr-Ag loading and to a lesser extent concentration dependent variation. If the material presents regions where the filler is not completely embedded into the matrix, or if pores appeared during polymerization or mixing, there will be voids that increase water absorption. The reduction in water absorption as a consequence of porosity decrease could be an indicator of a compact robust structure, with good mechanical strength. The decrease in water absorption could also be associated with the hydrophobic character of graphene. Moreover, high water absorption may deteriorate the mechanical properties of the dentures during their clinical use as a result of its plasticizing and degradation effects [39]. The color of the samples changed when adding the fillers, with a discreet difference for sample P1, but more evident for sample P2.

## 5. Conclusions

This study focused on the main mechanical properties, water absorption and morphology of PMMA samples loaded with 1% and respectively 2% Gr-Ag.

The practical implications of this study could be important for the clinical use of PMMA denture material; our results revealed a significantly enhanced mechanical behavior of the Gr-Ag enriched PMMA resin with respect to compression behavior, flexural profile and tensile strength. From the mechanical evaluation, we could conclude that 1% Gr-Ag content would be sufficient for the material to endure higher applied loads, to exhibit higher flexural strength and tensile characteristics compared to the unmodified PMMA. However, a content of 2% Gr-Ag showed lower ratios of absorbed water, which could reduce the risks of water mediated degradation effects.

## Figures and Tables

**Figure 1 materials-12-02335-f001:**
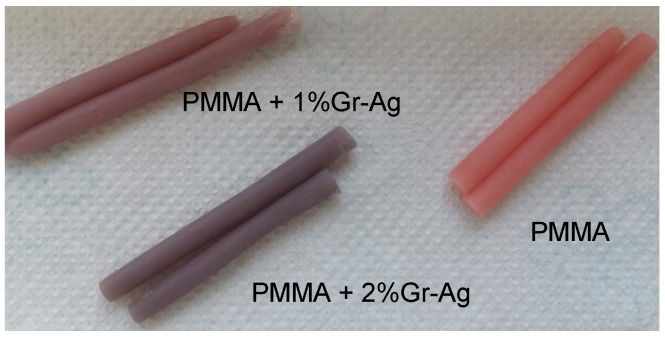
The obtained sticks after complete curing.

**Figure 2 materials-12-02335-f002:**
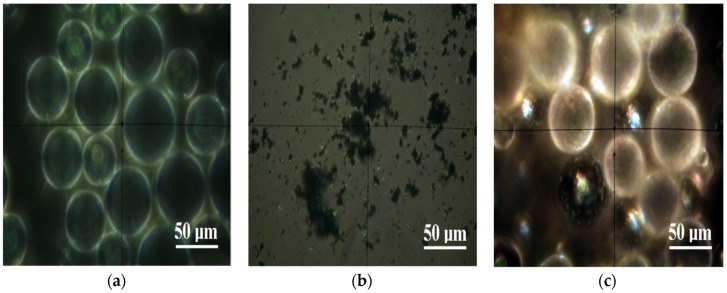
The microscopy in polarized light of the PMMA (**a**), Gr-Ag (**b**), and PMMA+1%Gr-Ag (**c**) powders, which are part of the composition of the materials investigated.

**Figure 3 materials-12-02335-f003:**
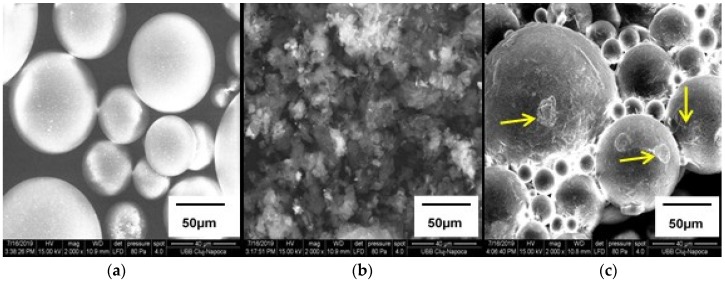
Surface images of fillers (**a**) PMMA, (**b**) Gr-Ag and (**c**) PMMA + 1% Gr-Ag captured with 2000× magnitude.

**Figure 4 materials-12-02335-f004:**
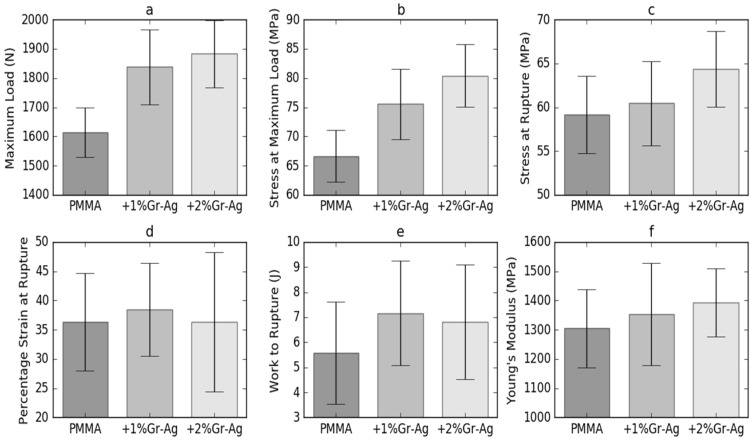
Compressive strength evaluation parameters and standard deviation bars. (**a**) Maximum load, (**b**) stress at maximum load, (**c**) stress at rupture, (**d**) strain at rupture, (**e**) work of rupture, (**f**) Young’s modulus.

**Figure 5 materials-12-02335-f005:**
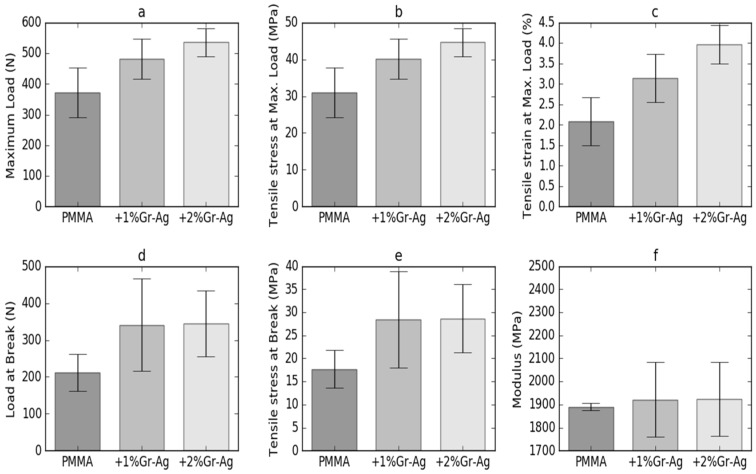
Tensile strength evaluation parameters and standard deviation bars. (**a**) Maximum load, (**b**) tensile stress at maximum load, (**c**) tensile strain at maximum load, (**d**) load at break, (**e**) tensile stress at break, (**f**) modulus.

**Figure 6 materials-12-02335-f006:**
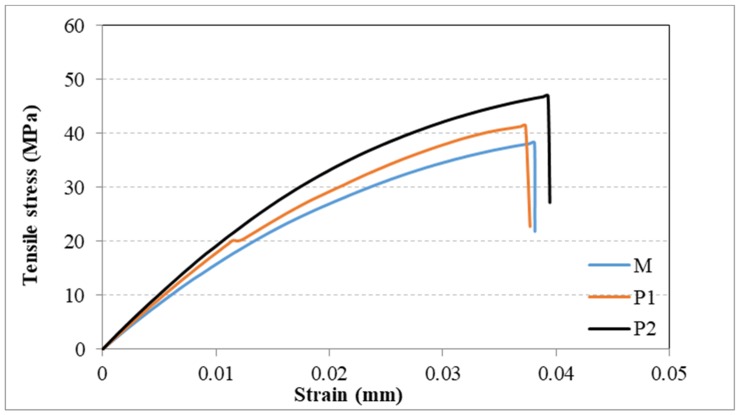
The experimental tensile stress vs. strain curves for samples M, PMMA sample; P1, PMMA + 1%Gr-Ag; P2, PMMA + 2%Gr-Ag.

**Figure 7 materials-12-02335-f007:**
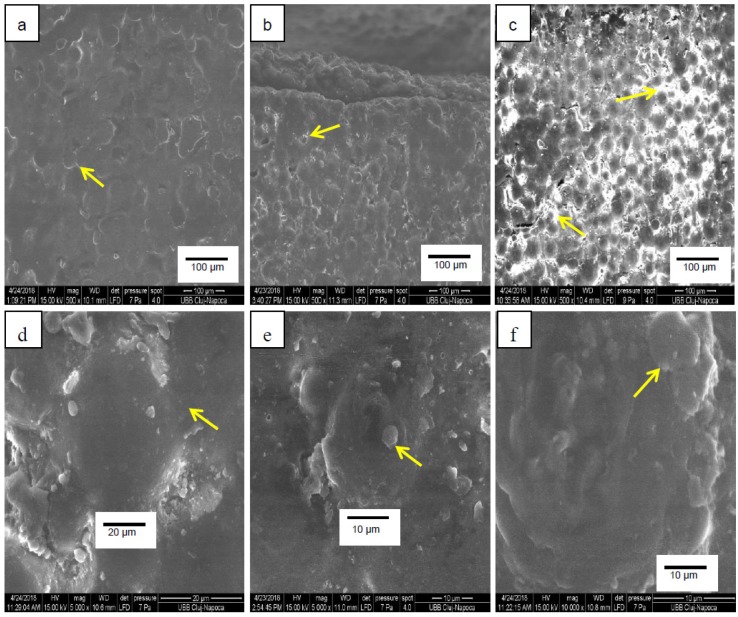
Surface images of samples (**a**) M (PMMA), (**b**) P1 (PMMA + 1% Gr-Ag) and (**c**) P2 (PMMA + 2% Gr-Ag) captured with 500× magnitude and surface images of samples (**d**) M (PMMA), (**e**) P1 (PMMA + 1% Gr-Ag) and (**f**) P2 (PMMA + 2% Gr-Ag) captured with 5000× magnitude.

**Figure 8 materials-12-02335-f008:**
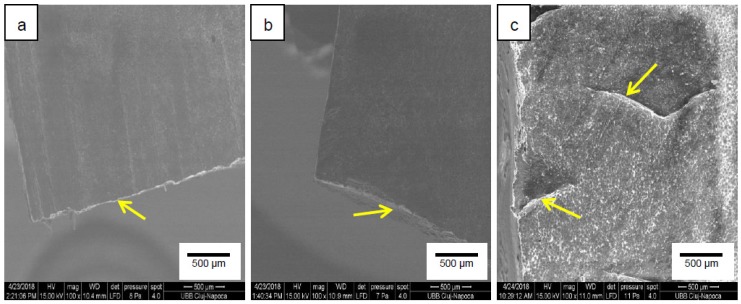
Fracture surface images of samples (**a**) M (PMMA), (**b**) P1 (PMMA + 1% Gr-Ag), and (**c**) P2 (PMMA + 2% Gr-Ag) captured with 100× magnitude.

**Table 1 materials-12-02335-t001:** Flexural strength evaluation parameters.

Sample	Maximum Load (N)	Modulus of Rupture (MPa)
M	18.1 ± 4.2 ^a^	13.4 ± 3.1 ^a^
P1	53.8 ± 19.2 ^b^	35.00 ± 8.3 ^b^
P2	81.3 ± 18.3 ^c^	36.89 ± 6.7 ^b^

M, PMMA sample; P1, PMMA + 1%Gr-Ag; P2, PMMA + 2%Gr-Ag. Different superscript letters indicate statistically significant variation (*p* < 0.05) between the samples.

**Table 2 materials-12-02335-t002:** Water absorption expressed as weight increase percentage after saliva exposure.

Sample	Saliva Exposure Time		Distilled Water Exposure Time	
	7 Days (%)	28 Days (%)	7 Days (%)	28 Days (%)
M	1.26 ± 0.13 ^a^	1.44 ± 0.07 ^a^	1.16 ± 0.19 ^a^	1.67 ± 0.64 ^a^
P1	0.78 ± 0.14 ^b^	1.21 ± 0.10 ^b^	1.25 ± 0.21 ^b^	1.65 ± 0.03 ^b^
P2	0.44 ± 0.10 ^c^	1.18 ± 0.34 ^b^	0.65 ± 0.15 ^c^	1.24 ± 0.15 ^b^

M(PMMA) sample; P1 (PMMA + 1%Gr-Ag); P2(PMMA + 2%Gr-Ag). Different superscript letters indicate statistically significant variation (*p* < 0.05) between the samples.

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
