# Peer review of "The Influence of Graphene in Improvement of Physico-Mechanical Properties in PMMA Denture Base Resins"

_materials, 2019, doi:10.3390/ma12142335_

Reviewer 1 Report

I have read the article “The influence of graphene in improvement of physico-mechanical properties in PMMA denture base resins” with genuine interest. Authors touch on the interesting problem of mechanical properties of polymeric fillers for dental applications. This remains an interesting area to pursue to study. However, their overall motivation is not explained sufficiently. The main idea is to use nanofillers containing some form of graphene, which is not at all specified, with antibacterial silver nanoparticles into nanocomposite nanofiller, described as GAgNPs. Subsequently, the Authors wanted to incorporate it in two different % (not described whether w/w or w/v) into PMMA polymeric matrix to look at the outcoming bulk material mechanical and degradation properties. At the moment – although interesting, the study lacks very basic information about the actual nanofillers used, including the exact synthesis route (just cited a paper of other Authors), what kind/type of graphene is used, its size, purity, number of layers, as well as what size of Ag NPs are used with no characteristics provided whatsoever. As indicated by many leading Authors in the field (e.g. Prof. Kostas Kostarelos), the exact characterisation of graphene materials remains crucial to understand their use in any applications, especially in biomedical fields. It is currently very hard to link the actual observations and increases in mechanical properties/water retention. The interactions between the nanofiller and matrix are not clear (what solvent was this done in and why? Is GAgNPs hydrophilic or hydrophobic?). Furthermore, these are not all linked to the actual needs of the field (i.e. what kind of degradation/mechanical properties would one want in dental fillings, what are current standards?). More exact and major points are listed below. I kindly suggest Authors to consider these points and undertake major revisions of the manuscript before it can be deemed publishable.

1.     The description of mechanical properties in the introduction is rather a summary of some literature chosen with no discussion of what the actual desired properties for the desired materials of interest should be. Can authors quote the desired mechanical and degradability characteristics that should be achieved in the field or are currently the best for used materials and improve their motivation with respect to the literature?

2.     The introduction to graphene and graphene-based materials is missing key references in the field, such as those by the Profs. Andre Geim and Kostas Novoselov. These should be included in the paragraph between lines 75-81. (e.g. Novoselov, K. S.; Geim, A. K.; Morozov, S. V.; Jiang, D.; Zhang, Y.; Dubonos, S. V.; Grigorieva, I. V.; Firsov, A. A., Electric field effect in atomically thin carbon films. Science 2004, 306 (5696), 666-669.; Geim, A. K.; Novoselov, K. S., The rise of graphene. Nature Materials 2007, 6 (3), 183-191.)

3.     Authors explain their motivation between the lines 82-87. With respect to this, I think its important to extend the motivation of study and:

a)     Relate the use of Silver NPs and their antibacterial properties to the needs in the field, i.e. improper use of dental implants/fillings without proper sterilisation is a cause of many post-operative infections. With the current use of antibiotics, one needs to provide alternatives, such as use of Ag NPs.

b)     Explain exactly whether the Authors aimed here at using Graphene Oxide, reduced Graphene Oxide or pure Graphene and the way they have achieved their materials (i.e. synthesis/preparation route).

c)     Explain their chosen ratios of graphene to NPs and the reason behind these chosen ratios (e.g. are chosen Ag NPs loading concentrations relating to those used in the literature for similar purposes?)

d)     Where did the Authors envisaged to have Ag NPs? Were they supposed to stay inside on the graphene or be dispersed in the polymer? How effective would you expect the Ag NPs bound to graphene to be in terms of their antibacterial properties if they are mixed inside the polymer matrix? Would you expect any release of these particles or would they be completely bound to graphene?

4.     It remains unclear of how and what Authors exactly mean by GAgNPs. In the materials and method section: “2.1 Graphene-based nanomaterials preparation” part they only provide reference [17] to the previously described method. Typically, in such cases one should follow that up by the additional description (i.e. Briefly, XYZ was done), especially if the previous method is not published by the same Authors. I would strongly suggest including this and quote the exact type of graphene produced and its ratios with Ag NPs. When Authors write 1 % - do they mean 1 w/w% or 1w/v% or any other %? Please clarify this and give information about how much of graphene and how much of Ag NPs are inside those 1 and 2 %.

5.     In 2.3 Water absorption method Authors suddenly mention that they are measuring the absorption of artificial saliva – this is an interesting point that has not been mentioned in the introduction and motivation. I could see for obvious reasons why this is relevant, and I would suggest including the explanation for it in the introduction/motivation.

6.     In the Materials and Methods section and at the beginning of Results Authors quote samples, P1, P2 and M. Reader won’t know what these are unless they are appropriately references to the table 1 that explains the used samples.

7.     I would suggest putting more effort into improving Figure 1 – increasing font sizes of both axis and making it neater. It is currently hard to read.

8.     The discussion of the mechanical properties does include some relevant references and pointing towards the increase of mechanical properties; however, it is currently not linked in any way to the actual needs of the field and the motivation. Does the 1% GAgNPs addition mean significant increase with comparison to what’s necessary for the dental fillers?  

9.     Lines 166-168: “The absorbed water results expressed as increase percentage after 7 and 28 days of water/saliva exposure are presented in table 2.” – the numbers given in Table 2 do not have % besides them. Does that mean that the weight has increased by 1.26% in 7 days for sample M or by 26% or by 126%? This needs clarification and proper units in the table.

10.  In my opinion, the SEM imaging so far does not prove the exact distribution of the used nanofiller in the polymer matrix. Authors do not provide any characteristics of the used nanofillers (GAgNPs). What kind of size of graphene was used (SEM of only nanofillers/AFM)? What purity (Raman spectroscopy)? How many layers (AFM)? What’s the size of Ag NPs (DLS)? The distribution of the nanofillers (GAgNPs) within PMMA matrix is crucial to understand the actual properties and this needs to be linked to the actual mechanical properties.

(The comments are also attached as .docx for convenience). 

Author Response

1.     Point 1. The description of mechanical properties in the introduction is rather a summary of some literature chosen with no discussion of what the actual desired properties for the desired materials of interest should be. Can authors quote the desired mechanical and degradability characteristics that should be achieved in the field or are currently the best for used materials and improve their motivation with respect to the literature?

Response 1. We completed with the mechanical and degradability characteristics that acrylic resin should be achieved.

 Point 2. The introduction to graphene and graphene-based materials is missing key references in the field, such as those by the Profs. Andre Geim and Kostas Novoselov. These should be included in the paragraph between lines 75-81. (e.g. Novoselov, K. S.; Geim, A. K.; Morozov, S. V.; Jiang, D.; Zhang, Y.; Dubonos, S. V.; Grigorieva, I. V.; Firsov, A. A., Electric field effect in atomically thin carbon films. Science 2004, 306 (5696), 666-669.; Geim, A. K.; Novoselov, K. S., The rise of graphene. Nature Materials 2007, 6 (3), 183-191.)

Response 2.  We mentioned in the text the reference recommended.

Point 3. Authors explain their motivation between the lines 82-87. With respect to this, I think its important to extend the motivation of study and:

Point 3.a)     Relate the use of Silver NPs and their antibacterial properties to the needs in the field, i.e. improper use of dental implants/fillings without proper sterilisation is a cause of many post-operative infections. With the current use of antibiotics, one needs to provide alternatives, such as use of Ag NPs.

Response 3.a. The antibacterial properties of silver nanoparticles at low concentration combined with the graphene reduce biofilm formation and bacterial and fungal colonization.

Point 3.b)     Explain exactly whether the Authors aimed here at using Graphene Oxide, reduced Graphene Oxide or pure Graphene and the way they have achieved their materials (i.e. synthesis/preparation route).

Response 3.b) The materials used is graphene with 6 layers decorated with Ag nanoclusters. The detalied synthesis are presented in the reference no. 20. At the materials and methods chapter we mentioned de AgNPs synthesis.

Point 3.c)     Explain their chosen ratios of graphene to NPs and the reason behind these chosen ratios (e.g. are chosen Ag NPs loading concentrations relating to those used in the literature for similar purposes?)

Response 3.c. We used Ag NPs because of their antibacterial properties, properties intensively studied in literature. The 3% of these in materials is the percent who gives better antibacterial properties and integration into graphene sheets.

d)    Point 3.d. Where did the Authors envisaged to have Ag NPs? Were they supposed to stay inside on the graphene or be dispersed in the polymer? How effective would you expect the Ag NPs bound to graphene to be in terms of their antibacterial properties if they are mixed inside the polymer matrix? Would you expect any release of these particles or would they be completely bound to graphene?

                    Response 3.d. AgNp are into the graphene sheets. We added in the graphene for an                         improvment of antibacterial properties of the material. We expect to release into the                             polymer matrix. We made antibacterial tests which demonstrate that adition in the polimer                     matrix of the GAgNPs improved antibacterial effect, compared with PMMA materials                         without GAgNp. These results will be published in a paper

Point 4.  It remains unclear of how and what Authors exactly mean by GAgNPs. In the materials and method section: “2.1 Graphene-based nanomaterials preparation” part they only provide reference [17] to the previously described method. Typically, in such cases one should follow that up by the additional description (i.e. Briefly, XYZ was done), especially if the previous method is not published by the same Authors. I would strongly suggest including this and quote the exact type of graphene produced and its ratios with Ag NPs. When Authors write 1 % - do they mean 1 w/w% or 1w/v% or any other %? Please clarify this and give information about how much of graphene and how much of Ag NPs are inside those 1 and 2 %.

Response 4. We completed the materials and methods section regarding the meaning of GAgNp. We clarified that 1, respectively 2 is wt%

5.     Point 5. In 2.3 Water absorption method Authors suddenly mention that they are measuring the absorption of artificial saliva – this is an interesting point that has not been mentioned in the introduction and motivation. I could see for obvious reasons why this is relevant, and I would suggest including the explanation for it in the introduction/motivation.

Response 5. We added in the introduction the motivation of studiing absorbtion in artificial saliva.

The dentures are used inside the oral cavity, in a wet environment that can lead to dimensional changes and affect the resin´s characteristics as water molecules enter between the polymeric chains and act as plasticizers, altering the physical and mechanical properties of the material; therefore water absorbtion of new denture base resins is important to be studied.

 6.     Point 6. In the Materials and Methods section and at the beginning of Results Authors quote samples, P1, P2 and M. Reader won’t know what these are unless they are appropriately references to the table 1 that explains the used samples.

Response 6. We specified what is the samples used and the composition of them.

7.  Point 7. I would suggest putting more effort into improving Figure 1 – increasing font sizes of both axis and making it neater. It is currently hard to read.

Response 7. We changed figure 1.

8.     Point 8. The discussion of the mechanical properties does include some relevant references and pointing towards the increase of mechanical properties; however, it is currently not linked in any way to the actual needs of the field and the motivation. Does the 1% GAgNPs addition mean significant increase with comparison to what’s necessary for the dental fillers? 

Response 8. We completed with the motivation of using these materials.

The resistance of the material to the stresses caused by mastication can be predicted by evaluating its mechanical properties.

From the mechanical evaluation, we could conclude that 1% GAgNP content would be sufficient for the material to endure higher applied loads, to exhibit higher flexural strength and tensile characteristics compared to the unmodified PMMA.

9.     Point 9. Lines 166-168: “The absorbed water results expressed as increase percentage after 7 and 28 days of water/saliva exposure are presented in table 2.” – the numbers given in Table 2 do not have % besides them. Does that mean that the weight has increased by 1.26% in 7 days for sample M or by 26% or by 126%? This needs clarification and proper units in the table.

Response 9. We added in the table the “%”, for a better understanding.

10Point 10.  In my opinion, the SEM imaging so far does not prove the exact distribution of the used nanofiller in the polymer matrix. Authors do not provide any characteristics of the used nanofillers (GAgNPs). What kind of size of graphene was used (SEM of only nanofillers/AFM)? What purity (Raman spectroscopy)? How many layers (AFM)? What’s the size of Ag NPs (DLS)? The distribution of the nanofillers (GAgNPs) within PMMA matrix is crucial to understand the actual properties and this needs to be linked to the actual mechanical properties.

Response 10. The working protocol is not provided for graphene characterization, only for PMMA materials with graphene. The characterisation of GAgNp is well described in reference 20.

Reviewer 2 Report

The Manuscript materials-535745 reports the preparation and mechanical characterization of graphene-PMMA composites for dental application. Two composites were prepared obtained by adding 1% or 2% of graphene doped with Ag nanoparticles into a commercially available self-curing PMMA resin. The composites were characterized by compressive, tensile and flexural tests as well as by water adsorption assays.

General comment

The study is quite simple but clearly presented and does not appear to have major technical flaws. The results are interesting under an applicative point of view. The weakness of the work is the lack of in vitro cytocompatibility study to assess possible toxicity of resins added with graphene and silver. Additionally, some important information on materials and methods are missing, such as the source of graphene and the method used to introduce silver (in this latter case authors only cited a reference).  

Specific comments:

1) Page 2 Section 2.1: Please add the source of graphene (in-house prepared?) and briefly describe the method used to add silver nanoparticles.

2) Results: All of the data have too much digits, many of which are not significant according to the related errors (e.g. 1614.34 ± 84.48 N). Please, rewrite the data all over the text.

3) Page 3 Section 3.1. “Compression”: The maximum load and the stress at rupture of the samples reported in Figure 1 have not the same trend. That suggests the presence of a Yield point in at least one of the samples. Therefore, it is useful to add a figure showing the compressive curves for all of the samples.

4)  Page 3 line 129: Change the word “same” with “similar”.

5) Figure 1: Add error bars in the histograms.

6) Figure 2: Add error bars in the histograms.

7) Page 4 lines 150-151. The sentence “A slight increase was al also shown for the Modulus, from 17.68 for M to 28.71 for P2” doesn’t seem coherent with the data reported in Fig.2f.

8) Page 5 line 155: I don’t see superscript letters in Figure 2.

9) Figure 3: Please report tensile stress vs. strain (%) and not strain (mm). The sentence at Page 5 lines 158-159 is not evident from the tensile curves reported in Figure 3.

10) Table 2: Water adsorption data obtained for immersion of samples in water are missing. Please add them and comment.

11) Discussion: Page 8 lines 216-232: The sample P1 has a odd behaviour that the authors should comment. The authors stated that addition of GAgNP increased the stress at rupture and decreased the strain at rupture. This is not true for P1.

12) Page 9 lines 263-266: I do not agree with the hypothesis on water adsorption decrease as a function of filler content increase (better filler distribution). I believe that graphene hydrophobicity affects water uptake of the composite. Of course, such influence is related to graphene content in the sample.  

Author Response

Point 1. Page 2 Section 2.1: Please add the source of graphene (in-house prepared?) and briefly describe the method used to add silver nanoparticles.

Response 1. We added the source of graphene and we described the method for adding silver nanoparticles.

 Point 2. Results: All of the data have too much digits, many of which are not significant according to the related errors (e.g. 1614.34 ± 84.48 N). Please, rewrite the data all over the text.

Response 2. We modified all data with one digit.

 Point 3. Page 3 Section 3.1. “Compression”: The maximum load and the stress at rupture of the samples reported in Figure 1 have not the same trend. That suggests the presence of a Yield point in at least one of the samples. Therefore, it is useful to add a figure showing the compressive curves for all of the samples.

Response 3. We modified by introducing error bars. It is possible that one of the samples presented a Yield point.

Point 4. Page 3 line 129: Change the word “same” with “similar”.

Response 4. We modified.

Point 5. Figure 1: Add error bars in the histograms.

Response 5. We modified.

Point 6. Figure 2: Add error bars in the histograms.

Response 6.  We added.

Point 7. Page 4 lines 150-151. The sentence “A slight increase was al also shown for the Modulus, from 17.68 for M to 28.71 for P2” doesn’t seem coherent with the data reported in Fig.2f.

Response 7. Done.

 Point 8. Page 5 line 155: I don’t see superscript letters in Figure 2.

Response 8. It is superscript letters in figure 2.

Point 9. Figure 3: Please report tensile stress vs. strain (%) and not strain (mm). The sentence at Page 5 lines 158-159 is not evident from the tensile curves reported in Figure 3.

Response 9. We modified the figure.

 Point 10. Table 2: Water adsorption data obtained for immersion of samples in water are missing. Please add them and comment.

Response 10. We completed with data for water adsorption.

Point 11. Discussion: Page 8 lines 216-232: The sample P1 has a odd behaviour that the authors should comment. The authors stated that addition of GAgNP increased the stress at rupture and decreased the strain at rupture. This is not true for P1.

Response 11. Indeed we foun that mistake. The correct statement is the addition of GAgNP for P1 increased both stress and strain at rupture (P2 increased the stress at rupture and decreased the strain at rupture).

Point 12. Page 9 lines 263-266: I do not agree with the hypothesis on water adsorption decrease as a function of filler content increase (better filler distribution). I believe that graphene hydrophobicity affects water uptake of the composite. Of course, such influence is related to graphene content in the sample. 

Response 12.  We agree that this could also be an explanation for the results we obtained.

We added it in the text. Line 311-312: The decrease in water absorption could also be associated with the hydrophobic character of graphene.

Author Response

Point 1. How many samples are tested? Error bars for each measured quantity should be included in Figure 1 and Figure 2 to make the measurement convincing.

Response 1. We tested 10 for each sample.

Point 2. There is a bump in the curve P1 in Figure 3, which means some slips between the sample and the gripper are happened in the experiments. This indicates the collected data is unreliable.

Response 2. We modified the figure.

Reviewer 4 Report

This study focuses on nanocomposite reinforcement with the present of Graphene and Ag in a blending system. Some physical properties were investigated such as mechanical deformation, strength and also water uptake. 

The following points need to be addressed by authors: 

- line 92: the samples P1 and P2 contained 1% and 2% GAgNP :1 and 2% of what? the unit and the reference mass of the measurement must be reported. 

- as the dispersing of the Nanoparticles including Graphite and its derivatives is still a dilemma. I strongly suggest rewriting the Graphene-based nanomaterials preparation section. If you want to cite somewhere, it's better to briefly explain the procedure.

- what is the specification of Graphene such as manufacturer, particle size, and so on?

- which method has been followed for statistical significance for all measurement in which the lack of error bars could question the accuracy of the data collections?

- due to what scientifical reason, their samples have been limited into two?

-I couldn't find characterisations for antimicrobial interactions of Ag or Gr

-1-2 wt.% of Graphene and AG needs to characterized if the color of the final nanocomposite is more likely to be changed?  

Author Response

Point 1: line 92: the samples P1 and P2 contained 1% and 2% GAgNP :1 and 2% of what? the unit and the reference mass of the measurement must be reported.

Response 1: We modified.

Point 2: as the dispersing of the Nanoparticles including Graphite and its derivatives is still a dilemma. I strongly suggest rewriting the Graphene-based nanomaterials preparation section. If you want to cite somewhere, it's better to briefly explain the procedure.

Response 2: We modified.

 Point 3: what is the specification of Graphene such as manufacturer, particle size, and so on?

Response 3: The graphene was obtained by our colleagues from Institute for Research and Development of Isotopic and Molecular Technologies, Romania. The detailes are presented in reference 20.

Point 4: which method has been followed for statistical significance for all measurement in which the lack of error bars could question the accuracy of the data collections?

Response 4: We used one-way ANOVA to test whether the mechanical characteristics of the three materials are different.

Point 5: due to what scientifical reason, their samples have been limited into two?

Response 5: Were made more samples, but were limited by the colour and the properties obtained.

Point 6: I couldn't find characterisations for antimicrobial interactions of Ag or Gr

Response 6: In this paper we didn’t investigate the antimicrobial interactions of Ag or Gr. These tests were done, but will be published in another paper.

Point 7: 1-2 wt.% of Graphene and AG needs to characterized if the color of the final nanocomposite is more likely to be changed? 

Response 7: The colour of the samples changed when adding the fillers, with a discreet difference for sample P1, but more evident for sample P2.

Round  2

Reviewer 1 Report

I have read the revised version of the manuscript and although I appreciate the Author’s efforts into revising this manuscript, I still believe that it needs to undergo major revisions (especially points from 6 onwards) before it can be deemed publishable. The list of the comments/replies to the responses is included here:

1)   For response 1: I still can’t find a ‘master’ mechanical properties in the introduction that would describe the desired materials. Can you provide quantitative data rather than qualitative description please?

2)   Line 58: Baker at al.  should be Baker et al.  

3)   Line 60: than the larger ones [7]. – please be specific at this point. What does large mean?

4)   Lines 70-73: The sentence is long and not understandable in English.

5)   Lines 108-111: by chemical chemical vapor deposition by radio frequency (RF-CCVD). – the proper name of the method is radio-frequency chemical vapor deposition. Please correct it throughout the manuscript.

6)   Line 111-112: The obtained graphene have 6 layers decorated with Ag nanoclusters and response 10 – I still believe that the Authors should provide actual data to prove this rather than only referencing source paper. The reproduction of the same materials is not as simple as repeating exact method from another work. To deem this work publishable – I would suggest at least including atomic force microscopy (AFM) scans of their produces nanofiller that shows a) thickness of the nanofiller as 6 layers and b) distribution and size of AgNPs in the nanocomposite. Raman spectroscopy could also be used to identify the quality of graphene along with the possible effect of surface enhancement that might show silver nanoparticles if the distribution is sufficient.

7)   With respect to Author’s response 3d: The Author’s say that they have included the antibacterial tests in the manuscript – this is currently not available in version 2. Also – they suggested that AgNPs would be released from graphene nanocomposite – can you provide any data/release assays to show that? Is the graphene -AgNPs acting as a delivery vehicle or is it a nanocomposite with strongly bound AgNPs that still persists to be antibacterial? I believe these tests should be included in the paper before the publication along with the full characterisation of the nanocomposite itself as per above point.

8)   Standard deviations in Fig 1 are overlapping between all measured samples significantly – what P values were obtained from statistical analysis between all sets using students (or otherwise statistic test)? Can authors indicate this on the actual graph with a * comparison, as per usual standards?

9)   Error bars are missing in Fig 2. Same comment as above.

10)           Fig 3 is not of sufficient quality to merit publication. I would kindly urge authors to revise the Fig. to represent the data and scales better, in a similar way to what they had previously and to the new Figures 1 and 2.

11)           Does the fact that PMMA contain acrylic groups disallow it to be used as any biomedical device due to present acrylic groups that have been shown to be strongly linked to cancer case? Can authors please comment on that in the paper?

12)           Lines 312-313: 36]. The colour of the samples changed when adding the fillers, with a discreet difference 312 for sample P1, but more evident for sample P2. – can authors provide optical photographs of these samples for comparison? (could be included as supporting information).

Author Response

 Point 1. For response 1: I still can’t find a ‘master’ mechanical properties in the introduction that would describe the desired materials. Can you provide quantitative data rather than qualitative description please?

Response 1. We done.

Point 2. Line 58: Baker at al.  should be Baker et al.  

Response 2. We modified.

Point 3.  Line 60: than the larger ones [7]. – please be specific at this point. What does large mean?Response 3. We changed.

Point 4. Lines 70-73: The sentence is long and not understandable in English.

Response 4. We modified.

Point 5.  Lines 108-111: by chemical chemical vapor deposition by radio frequency (RF-CCVD). – the proper name of the method is radio-frequency chemical vapor deposition. Please correct it throughout the manuscript.

Response 5. We modified.

Point 6.  Line 111-112: The obtained graphene have 6 layers decorated with Ag nanoclusters and response 10 – I still believe that the Authors should provide actual data to prove this rather than only referencing source paper. The reproduction of the same materials is not as simple as repeating exact method from another work. To deem this work publishable – I would suggest at least including atomic force microscopy (AFM) scans of their produces nanofiller that shows a) thickness of the nanofiller as 6 layers and b) distribution and size of AgNPs in the nanocomposite. Raman spectroscopy could also be used to identify the quality of graphene along with the possible effect of surface enhancement that might show silver nanoparticles if the distribution is sufficient.

Response 6. The protocol did not propose the synthesis and characterization of the graphene, the aim was to use them in a small percentage of the material used in the dental prosthesis. The purpose of the paper is the influence of graphene on the materials used in dental prosthetics in terms of physico-mechanical properties, purpose achieved and presented in the paper. The characterization of the graphene was made and published by the INCDTIM team.

Point 7. With respect to Author’s response 3d: The Author’s say that they have included the antibacterial tests in the manuscript – this is currently not available in version 2. Also – they suggested that AgNPs would be released from graphene nanocomposite – can you provide any data/release assays to show that? Is the graphene -AgNPs acting as a delivery vehicle or is it a nanocomposite with strongly bound AgNPs that still persists to be antibacterial? I believe these tests should be included in the paper before the publication along with the full characterisation of the nanocomposite itself as per above point.

Response 7. The antibacterial effect studied on the materials presented in this paper were included in a paper, which are under review.

Point 8.    Standard deviations in Fig 1 are overlapping between all measured samples significantly – what P values were obtained from statistical analysis between all sets using students (or otherwise statistic test)? Can authors indicate this on the actual graph with a * comparison, as per usual standards?

Response 8. The statistical significance of the differences for the 10 samples for each of the three materials was tested using one-way Anova test (Microsoft Office, Excel).

Point 9. Error bars are missing in Fig 2. Same comment as above.

Response 9. We modified the figure.

Point 10. Fig 3 is not of sufficient quality to merit publication. I would kindly urge authors to revise the Fig. to represent the data and scales better, in a similar way to what they had previously and to the new Figures 1 and 2.

Response 10. We modified.

Point 11. Does the fact that PMMA contain acrylic groups disallow it to be used as any biomedical device due to present acrylic groups that have been shown to be strongly linked to cancer case? Can authors please comment on that in the paper?

Response 11. We done and we added a reference 35.

Point 12. Lines 312-313: 36]. The colour of the samples changed when adding the fillers, with a discreet difference 312 for sample P1, but more evident for sample P2. – can authors provide optical photographs of these samples for comparison? (could be included as supporting information).

Response 12. We added an optical photographs of the samples.

Reviewer 2 Report

The authors did not reply to two suggestions. Specifically, compression curves were not shown in the text and error bars in the histograms were not added in  figure 2.

Author Response

Point 1. The authors did not reply to two suggestions. Specifically, compression curves were not shown in the text and error bars in the histograms were not added in figure 2.

Response 1.  We added.

Reviewer 4 Report

They have done a good attempt to clear the research outlines however I think the following points should be reconsidered to make the work publishable. 

- The first sentence of the abstract is better to be changed when the study of long term deformation (as a function of time) and wear resistance (as a function of the use cycle) is not central points of their research. 

-The claims have to be addressed scientifically in research articles, for instance:  line 98, page 2: improving the antimicrobial denture composite properties as a part of their hypothesis has been claimed while no evaluation can be observed. I suggest if the authors wanted to investigate the mechanical properties and water uptake of the nanocomposite, the other none-considered assumptions amended in their hypothesis. 

-  It seems that they have used a standard for mechanical tests, I suggest to name the standard which credit the reliability of the tests. 

- In figure 1, Compressive strength evaluation, are the bars representing the error or standard deviation, in the caption they need to be introduced.  

- Tensile strength evaluation also needs standard deviation bars in figures. 

- GAgNP has been referred by different names that need to be the same. I think Gr-Ag is more appropriate for this compound, as it has been mentioned by the authors in the material section.  

- Again I couldn't find the amount of  Graphene and silver individually in Gr-Ag compound which is highly important to be introduced in their material section; otherwise, the reproducibility of the work and correctness of data can be questioned by readers.

Author Response

Point 1.  The first sentence of the abstract is better to be changed when the study of long term deformation (as a function of time) and wear resistance (as a function of the use cycle) is not central points of their research. 

Response 1. We modified.

Point 2. The claims have to be addressed scientifically in research articles, for instance:  line 98, page 2: improving the antimicrobial denture composite properties as a part of their hypothesis has been claimed while no evaluation can be observed. I suggest if the authors wanted to investigate the mechanical properties and water uptake of the nanocomposite, the other none-considered assumptions amended in their hypothesis. 

Response 2. We changed.

Point 3. It seems that they have used a standard for mechanical tests, I suggest to name the standard which credit the reliability of the tests. 

Response 3. The samples for the mechanical evaluation of compressive strength (CS), diametral tensile strength (DTS) and flexural strength (FS) were prepared according to ISO 4049/2000 and international norms “American Dental Association’s Specification” No. 27.

Point 4. In figure 1, Compressive strength evaluation, are the bars representing the error or standard deviation, in the caption they need to be introduced.  

Response 4. We added.

Point 5. Tensile strength evaluation also needs standard deviation bars in figures. 

Response 5. We added.

Point 6. GAgNP has been referred by different names that need to be the same. I think Gr-Ag is more appropriate for this compound, as it has been mentioned by the authors in the material section.  

Response 6. We changed in all the manuscript.

Point 7. Again I couldn't find the amount of Graphene and silver individually in Gr-Ag compound which is highly important to be introduced in their material section; otherwise, the reproducibility of the work and correctness of data can be questioned by readers.

Response 7. The amount of Gr-Ag in the compound is 3%, it’s specified at chapter 2.1.

Round  3

Reviewer 1 Report

Point 1: (relating to response 1)

Can Author’s quote the exact mechanical properties values in lines 43-46 ( Apparently, close to 70% of dentures brake in the first three years of wearing due to the repeated masticatory forces that cause material fatigue or as a result of prosthesis dropping. Moreover, the stresses developed in the oral cavity during biting and mastication have a deforming effect which can initiate fractures in time [4].) when describing the current state-of-the art technology? This could really help understand comparison of values (Young’s modulus, tensile stress, break etc.) to the values obtained in the presented work.

Point 2:

Lines 56-58: Baker et al. assessed the antibacterial properties of silver nanoparticles at low concentrations, both in solution and on Petri dishes, demonstrating that smaller particles can be more efficient against bacteria due to their larger surface to volume ratio [7].

Can Author’s quote exact size of nanoparticles used in this study in this sentence? If the reader came not from the field, they would still not know what kind of Ag NPs are required (in terms of size) and could not compare to the actual particles used by the Authors in the work.

Point 3: (relating to last version point 6/Response 6)

Response 6. The protocol did not propose the synthesis and characterization of the graphene, the aim was to use them in a small percentage of the material used in the dental prosthesis. The purpose of the paper is the influence of graphene on the materials used in dental prosthetics in terms of physico-mechanical properties, purpose achieved and presented in the paper. The characterization of the graphene was made and published by the INCDTIM team.

Although I appreciate Author’s response, I still strongly believe that this work requires presenting the actual data describing nanocomposite fillers used here. Even if the synthesis of the material was already published by others in the field (irrespectively of whether these are Author’s collaborators or not), every follow-up article requires the full description of used materials (re-synthesized), with the data presented at least in the Supporting Information. This is really important to retain the reproducibility of the results by any others in the field. I strongly encourage Author’s to fill in the gaps by providing Supporting Information about the nanocomposite fillers used in this study to complete their story and give the full picture of their study. The size of used graphene, of used Ag nanoparticles and the distribution of Ag NPs on the graphene will be crucial factors affecting the final bulk material properties.

Point 4: (relating to response 7)

Response 7. The antibacterial effect studied on the materials presented in this paper were included in a paper, which are under review.

Unfortunately I can’t understand Author’s motivation to split the work into 2 publications.

I still believe that the additional antibacterial tests should be included in the paper before the publication; especially that Author’s motivation remained to use antibacterial nanocomposite filler. Otherwise why not only use Graphene? See point 5 for follow-up.

Point 5:

Why would one use Gr-Ag at all to increase the mechanical properties if only Gr could be used? Actually – can Author’s present any mechanical properties control experiments with only Graphene or only Silver Nanoparticles to show whether Gr-Ag has unique capability to change the bulk materials properties or perhaps if only one of the components (most likely Graphene) are sufficient?

Overall I believe that Author’s would still require answering points 3-5 to merit this work publishable.  Please take these points with careful consideration.

Author Response

Point 1: (relating to response 1)

Can Author’s quote the exact mechanical properties values in lines 43-46 ( Apparently, close to 70% of dentures brake in the first three years of wearing due to the repeated masticatory forces that cause material fatigue or as a result of prosthesis dropping. Moreover, the stresses developed in the oral cavity during biting and mastication have a deforming effect which can initiate fractures in time [4].) when describing the current state-of-the art technology? This could really help understand comparison of values (Young’s modulus, tensile stress, break etc.) to the values obtained in the presented work.

Response 1. Flexural strength Castavaria =79 Mpa (www.vertex-dental.com)   Tensile strength PMMA =76 MPa.

Point 2: Lines 56-58: Baker et al. assessed the antibacterial properties of silver nanoparticles at low concentrations, both in solution and on Petri dishes, demonstrating that smaller particles can be more efficient against bacteria due to their larger surface to volume ratio [7].

 Can Author’s quote exact size of nanoparticles used in this study in this sentence? If the reader came not from the field, they would still not know what kind of Ag NPs are required (in terms of size) and could not compare to the actual particles used by the Authors in the work.

Am incercat sa facem o analiza a pulberilor , unde s-a evidentiat prezenta Ag. Am introdus analiza prin microscopica electronica de baleiaj si optica a pulberilor investigate la punctele ….

Analiza pulberilor utilizate nu a fost obiectul acestui studiu, pentru ca s-au utilizat pulberi complet analizate.

Response 2. We modified.

Point 3: (relating to last version point 6/Response 6)

Response 6. The protocol did not propose the synthesis and characterization of the graphene, the aim was to use them in a small percentage of the material used in the dental prosthesis. The purpose of the paper is the influence of graphene on the materials used in dental prosthetics in terms of physico-mechanical properties, purpose achieved and presented in the paper. The characterization of the graphene was made and published by the INCDTIM team.

Although I appreciate Author’s response, I still strongly believe that this work requires presenting the actual data describing nanocomposite fillers used here. Even if the synthesis of the material was already published by others in the field (irrespectively of whether these are Author’s collaborators or not), every follow-up article requires the full description of used materials (re-synthesized), with the data presented at least in the Supporting Information. This is really important to retain the reproducibility of the results by any others in the field. I strongly encourage Author’s to fill in the gaps by providing Supporting Information about the nanocomposite fillers used in this study to complete their story and give the full picture of their study. The size of used graphene, of used Ag nanoparticles and the distribution of Ag NPs on the graphene will be crucial factors affecting the final bulk material properties.

Response 3. We introduced the scanning electron microscopy and the optical analysis of the powders investigated at the points 3.1 and 3.2.

Point 4: (relating to response 7)

Response 7. The antibacterial effect studied on the materials presented in this paper were included in a paper, which are under review.

Unfortunately I can’t understand Author’s motivation to split the work into 2 publications.

I still believe that the additional antibacterial tests should be included in the paper before the publication; especially that Author’s motivation remained to use antibacterial nanocomposite filler. Otherwise why not only use Graphene? See point 5 for follow-up.

Response 4. We performed extensive analysis of the antimicrobial properties of the studied materials using 6 different microbial strains, using 2 methods of analysis. As the study needed extensive description/comments + many figures, we decided to organize the material in another article that is currently under review.

Point 5: Why would one use Gr-Ag at all to increase the mechanical properties if only Gr could be used? Actually – can Author’s present any mechanical properties control experiments with only Graphene or only Silver Nanoparticles to show whether Gr-Ag has unique capability to change the bulk materials properties or perhaps if only one of the components (most likely Graphene) are sufficient?

Overall I believe that Author’s would still require answering points 3-5 to merit this work publishable.  Please take these points with careful consideration.

Response 5. Although AgNp proved to exhibit antibacterial effects, a decrease in the mechanical properties was reported by different authors, so we decided to add graphene in the attempt to improve the mechanical properties as well.

We added 2 references 35 and 36.

Reviewer 4 Report

The authors have modified the typos appropriately and answered the points that have mentioned as concerns in the previous reviews.  

Author Response

Thank you for your appreciation.